# Neuroligin-3-Mediated Synapse Formation Strengthens Interactions between Hippocampus and Barrel Cortex in Associative Memory

**DOI:** 10.3390/ijms25020711

**Published:** 2024-01-05

**Authors:** Huajuan Xiao, Yang Xu, Shan Cui, Jin-Hui Wang

**Affiliations:** 1Sino-Danish College, University of Chinese Academy of Sciences, Beijing 100049, China; huajuan.xiao18@gmail.com; 2College of Life Science, University of Chinese Academy of Sciences, Beijing 100049, China; xuyang186@mail.ucas.ac.cn; 3Institute of Biophysics, Chinese Academy of Sciences, Beijing 100101, China; cuishan@moon.ibp.ac.cn

**Keywords:** associative memory, hippocampus, barrel cortex, synapse connection, axon projection

## Abstract

Memory traces are believed to be broadly allocated in cerebral cortices and the hippocampus. Mutual synapse innervations among these brain areas are presumably formed in associative memory. In the present study, we have used neuronal tracing by pAAV-carried fluorescent proteins and *neuroligin-3* mRNA knockdown by shRNAs to examine the role of neuroligin-3-mediated synapse formation in the interconnection between primary associative memory cells in the sensory cortices and secondary associative memory cells in the hippocampus during the acquisition and memory of associated signals. Our studies show that mutual synapse innervations between the barrel cortex and the hippocampal CA3 region emerge and are upregulated after the memories of associated whisker and odor signals come into view. These synapse interconnections are downregulated by a knockdown of neuroligin-3-mediated synapse linkages. New synapse interconnections and the strengthening of these interconnections appear to endorse the belief in an interaction between the hippocampus and sensory cortices for memory consolidation.

## 1. Introduction

Associative memory is essential for cognitive activities and emotional reactions [1,2,3,4,5,6,7]. Memory traces, or engrams, for the storing of specific signals, are widely allocated in the cerebral brain, which is supported by numerous experimental data. For instance, the ablation of different areas of the cerebral brain did not completely remove rodents’ memory of a maze [8,9,10]. Memory-relevant engrams have been morphologically and functionally detected in the cerebral cortices, the hippocampus, and the amygdala [5,6,11,12,13,14,15,16,17,18]. However, how basic units in the memory trace work together to encode the joint storage and reciprocal retrieval of the specific associated signals remains largely unknown.

After the temporal lobe of cerebral cortices including the hippocampus was dissected from the cerebral brain in epileptic patients, the patients showed anterograde and retrograde amnesia [19,20,21,22,23,24,25]; thus, the hippocampus was thought to play a central role in memory formation [26,27]. This proposal has been supported by studies conducted on the hippocampus relevant to learning and memory [28,29,30]. The overexpression of some genes in the hippocampus increased the learning efficiency and memory consolidation in spatial location [31,32,33,34,35,36,37], while the knock-out of these genes led to a deficit of spatial memory [38,39,40,41]. The hippocampus appears essential for spatial learning and memory. However, new synapse interconnections and synapse-mediated interactions between the hippocampus and other cortical areas for learning and memory remain to be studied [6]. Molecular mechanisms for their interconnections and interactions, such as the effect of neuroligin-3, one of the proteins for synapse linkage [42,43,44,45,46,47,48], have to be elucidated for in vivo synapse interconnections.

In terms of memories of specific signals, or memory specificity, one line of studies focuses on elucidating the role of sensory cortices in associative learning and memory. These studies are based on the fact that different signals during learning are carried into the brain for storage by specific sensory systems [6]. The associations of the whisker tactile signal and the odorant signal in mice recruited their barrel and piriform cortical neurons to be associative memory cells, which were recruited to be their mutual synapse innervations and to encode these signals associatively for their integrative storage and reciprocal retrieval [16,49,50,51,52,53,54,55]. These primary associative memory cells in the sensory cortices projected their axons to the prefrontal cortex, the hippocampus, and other brain areas, and then, synaptically innervated the neurons that were recruited as secondary associative memory cells [6,56,57]. Whether these different grades of associative memory neurons are mutually innervated to endorse their interactions for memory formation and consolidation remains to be elucidated [6].

Taking the questions above, we intend to investigate the role of neuroligin-3-mediated synapse formation in the interconnection between the barrel cortex and the hippocampus during associative memory. The strategies we adopted to address this question are as follows. The pairing of a whisker tactile signal and an odorant signal was used as a mouse model of associative learning and memory [6,16]. The new formation and change of synapse interconnections between the barrel cortex and the hippocampus were examined by neural tracing, wherein, retrograde adeno-associated viruses (AAVs) carrying fluorescent protein genes were microinjected into either the hippocampal CA3 area or the barrel cortex, and subsequently, the fluorescent proteins expressed in neuronal somata were detected in the barrel cortex or hippocampal CA3/CA1 areas. The role of neuroligin-3-mediated synapse linkage in new synapse formation was studied by injecting short-hairpin RNA (shRNA) specific for *neurologin-3* into these areas, by which the synapse interconnections between the hippocampus and the barrel cortex were analyzed.

## 2. Results

### 2.1. The Connection from the Hippocampus to the Barrel Cortex Is Strengthened after the Formation of Associative Memory

The feedback action from the hippocampus to the barrel cortex was studied by detecting the connections from hippocampal neurons to barrel cortical neurons using AAV2/retro-CMV-EGFP injected into the barrel cortices (white arrows in Figure 1C) of PSG mice (bottom panels in Figure 1B,C) and UPSG mice (top panels in Figure 1B,C). The retrograde transportation from the axonal terminals of barrel cortical neurons to their somata was examined by scanning the hippocampal CA3 and CA1 areas under a confocal microscope. The fluorescent intensities in the hippocampal CA1 and CA3 areas from PSG mice appear higher than in those from UPSG mice. The relative fluorescent strengths of EGFP-labeled neurons in the hippocampal CA1 area at anterior–posterior (AP)-1.34 mm were 0.71 ± 0.17 in PSG mice (red bar in Figure 1D, *n* = 5) and 0.07 ± 0.01 in UPSG mice (blue bar, *n* = 5, *p* = 0.0079). The relative fluorescent strengths of EGFP-labeled neurons in the hippocampal CA3 area at AP-1.34 mm were 0.38 ± 0.17 in PSG mice (red bar in Figure 1D, *n* = 5) and 0.03 ± 0.01 in UPSG mice (blue bar, *n* = 5, *p* = 0.0079). Furthermore, the relative fluorescent strengths of EGFP-labeled neurons in the hippocampal CA1 area at AP-1.70 mm were 0.86 ± 0.43 in PSG mice (red bar in Figure 1E, *n* = 5) and 0.11 ± 0.03 in UPSG mice (blue bars, *n* = 5, *p* = 0.0079). The relative fluorescent strengths of EGFP-labeled neurons in the hippocampal CA3 area at AP-1.70 mm were 0.33 ± 0.11 in PSG mice (red bar in Figure 1E, *n* = 5) and 0.02 ± 0.01 in UPSG mice (blue bar, *n* = 5, *p* = 0.0079).

Confocal scanning was also conducted and analyzed under high magnification. The number and activity strength of EGFP-labeled hippocampal CA3 and CA1 neurons appear substantially raised in PSG mice (bottom panels in Figure 2A) in comparison with those in UPSG mice (top panels). The quantities of EGFP-labeled neurons versus total neurons (percentage) in the hippocampal CA1 area were 57.38 ± 2.60 in PSG mice (red bar in Figure 2B, *n* = 5) and 23.73 ± 2.26 in UPSG mice (blue bar, *n* = 5, *p* = 0.0079). The percentages of EGFP-labeled neurons in the hippocampal CA3 area were 63.44 ± 4.02 in PSG mice (red bar in Figure 2B, *n* = 8) and 32.93 ± 3.51 in UPSG mice (blue bar, *n* = 5, *p* = 0.0040). Moreover, fluorescent intensities in EGFP-labeled hippocampal CA1 neurons were 15.85 ± 0.66 × 10^2^ in PSG mice (red bar in Figure 2C, *n* = 104 cells from five mice) and 9.38 ± 0.36 × 10^2^ in UPSG mice (blue bars, *n* = 55 cells from five mice, *p* < 0.0001). Fluorescent intensities within EGFP-labeled hippocampal CA3 neurons were 14.96 ± 0.37 × 10^2^ in PSG mice (red bar in Figure 2C, *n* = 325 cells from five mice) and 8.46 ± 0.51 × 10^2^ in UPSG mice (blue bar, *n* = 74 cells from five mice, *p* < 0.0001). The results indicate that more hippocampal CA3 and CA1 neurons project their axons to barrel cortical neurons, and the activity strength of these hippocampal neurons significantly rises, after associative memory forms. These new axon projections from active hippocampal CA3 and CA1 neurons to barrel cortical neurons are linked to associative memory formation. This result endorses the hypothesis that secondary associative memory cells in hippocampal CA3/CA1 areas project their axons to innervate primary associative memory cells in the barrel cortex [6].

### 2.2. The Connections from the Barrel Cortex and Hippocampal CA1 Area to the Hippocampal CA3 Area Are Upregulated after the Formation of Associative Memory

We also examined the feedforward action by the connection from barrel cortical neurons to hippocampal CA1–3 neurons during associative memory using AAV2/retro-CMV-EGFP microinjected into the hippocampal CA3 area (white arrows in Figure 3B) in PSG mice (bottom panels in Figure 3A,B) and UPSG mice (top panels in Figure 3A,B). The retrograde transportation from the axon terminals of hippocampal CA3 neurons to their somata was examined by scanning the barrel cortices and hippocampal CA1 under a confocal microscope. Fluorescent intensities in barrel cortices and hippocampal CA1 from PSG mice appear higher than in those from UPSG mice. The relative fluorescent strengths of EGFP-labeled neurons in the barrel cortex at AP-1.70 mm were 0.11 ± 0.04 in PSG mice (red bar in Figure 3C, *n* = 5) and 0.01 ± 0.001 in UPSG mice (blue bar, *n* = 6, *p* = 0.0043). The relative strengths of EGFP-labeled neurons in the hippocampal CA1 area at AP-1.70 mm were 5.25 ± 0.45 in PSG mice (red bar in Figure 3D, *n* = 5) and 0.33 ± 0.03 in UPSG mice (blue bars, *n* = 6, *p* = 0.0043).

Confocal scanning was also conducted and analyzed under high magnification. The number and activity strengths of EGFP-labeled hippocampal CA1 neurons and barrel cortical neurons appear increased substantially in PSG mice (bottom panels in Figure 4A), in comparison with those in UPSG mice (top panels). The percentages of EGFP-labeled neurons in the barrel cortex were 65.42 ± 4.89 in PSG mice (red bar in Figure 4B, *n* = 5) and 2.27 ± 2.27 in UPSG mice (blue bar, *n* = 4, *p* = 0.0159). The percentages of EGFP-labeled neurons in the hippocampal CA1 area were 57.74 ± 6.47 in PSG mice (red bar in Figure 4B, *n* = 4) and 9.84 ± 2.79 in UPSG mice (blue bar, *n* = 6, *p* = 0.0238). Furthermore, fluorescent intensities in EGFP-labeled barrel cortical neurons were 9.53 ± 1.18 × 10^2^ in PSG mice (red bar in Figure 4C, *n* = 48 cells from five mice) and 1.8 ± 0.91 × 10^2^ in UPSG mice (blue bars, *n* = 5 from four mice, *p* = 0.0027). Fluorescent intensities within EGFP-labeled hippocampal CA1 neurons were 8.88 ± 0.52 × 10^2^ in PSG mice (red bar in Figure 4C, *n* = 45 cells from three mice) and 1.35 ± 0.16 × 10^2^ in UPSG mice (blue bar, *n* = 10 cells from 6 mice, *p* < 0.0001). These results indicate that more barrel cortical neurons project their axons to hippocampal CA3 neurons and that the activity strength of these barrel cortical neurons rises significantly after associative memory forms. The new axon projections from active barrel cortical neurons to hippocampal CA3 neurons are linked to associative memory formation. This result endorses the hypothesis that the primary associative memory cells in the barrel cortex project their axons to innervate the secondary associative memory cells in the hippocampus [6].

### 2.3. The Knockdown of Neuroligin-3 in the Barrel Cortex Attenuates the Connection from the Hippocampus to the Barrel Cortex

The data presented above support the view that associative learning and memory are based on new axon projections and synapse innervations by the coactivity among neurons, or their activity and interconnection [6]. To examine the causal relationship between new synapse interconnection between the sensory cortices and the hippocampal CA3 area for associative memory as well as its molecular mechanisms, we utilized the approach of knocking down the molecules relevant to synapse formation and linkage, such as neuroligin-3, one of the synapse linkage proteins [42,43,44,45,46]. Neuroligin-3 knockdown was conducted by shRNA specifically for *neuroligin-3* carried by adeno-associated viruses (AAVs) and microinjected in cortical areas [58,59,60,61,62]. The microinjections of AAV-DJ/8-U6-mNlgn3-GFP [pAAV(shRNA)-U6-mNlgn3-EGFP] into the barrel cortex or hippocampal CA3 region were conducted two days before the paradigm of associative learning. After the paradigm of associative learning, the mice in the subgroups of *neuroligin-3* knockdown and shRNA-scramble control were examined in terms of their synapse innervations on the hippocampus and the barrel cortex.

In the first set of experiments, we microinjected AAV-DJ/8-U6-mNlgn3-EGFP used to knockdown *neuroligin-3* and AAV2/retro-CMV-EGFP used for retrograde neural tracing (white arrow in bottom panel in Figure 5B) as well as AAV-DJ/8-U6-mNlgn3-scramble-EGFP for shRNA scramble control and AAV2/retro-CMV-EGFP for retrograde neural tracing into the barrel cortices (white arrow in top panel in Figure 5B). Fluorescent intensities in hippocampal CA1–3 areas from *neuroligin-3* knockdown in the barrel cortex appear lower than in those from *neuroligin-3* scramble control. The relative fluorescent strengths of EGFP-labeled neurons in the hippocampal CA1 area at AP-1.34 mm were 0.52 ± 0.08 in shRNA-scramble control mice (red bar in Figure 5C, *n* = 6) and 0.14 ± 0.03 in *neuroligin-3* knockdown mice (blue bar, *n* = 6, *p* = 0.0022). The relative fluorescent strengths of EGFP-labeled neurons in the hippocampal CA3 area at AP-1.34 mm were 0.11 ± 0.02 in shRNA scramble control mice (red bar in Figure 5C, *n* = 6) and 0.04 ± 0.00 in *neuroligin-3* knockdown mice (blue bar, *n* = 6, *p* = 0.0022). Furthermore, the relative fluorescent strengths of EGFP-labeled neurons in the hippocampal CA1 area at AP-1.7 mm were 0.72 ± 0.2 in shRNA scramble control mice (red bar in Figure 5D, *n* = 6) and 0.22 ± 0.05 in *neuroligin-3* knockdown mice (blue bars, *n* = 6, *p* = 0.0260). The relative fluorescent strengths of EGFP-labeled neurons in the hippocampal CA3 area at AP-1.70 mm were 0.12 ± 0.03 in shRNA scramble control mice (red bar in Figure 5D, *n* = 6) and 0.03 ± 0.01 in *neuroligin-3* knockdown mice (blue bar, *n* = 5, *p* = 0.0087).

Confocal images under high magnification were also captured and analyzed. The number and activity strength of EGFP-labeled hippocampal CA3 and CA1 neurons appear lower in *neuroligin-3* knockdown mice (bottom panels in Figure 6A), in comparison with those in shRNA scramble control mice (top panels). The percentages of EGFP-labeled neurons in the hippocampal CA1 area were 63.62 ± 4.47 in shRNA scramble control mice (red bar in Figure 6B, *n* = 6) and 39.86 ± 2.3 in *neuroligin-3* knockdown mice (blue bar, *n* = 6, *p* = 0.0006). The percentages of EGFP-labeled neurons in the hippocampal CA3 area were 68.89 ± 10 in shRNA scramble control mice (red bar in Figure 6B, *n* = 4) and 36.9 ± 4.23 in *neuroligin-3* knockdown mice (blue bar, *n* = 4, *p* = 0.0286). Moreover, fluorescent intensities in EGFP-labeled hippocampal CA1 neurons were 31.67 ± 0.65 × 10^2^ in shRNA scramble control mice (red bar in Figure 6C, *n* = 171 cells from six mice) and 20.14 ± 0.89 × 10^2^ in *neuroligin-3* knockdown mice (blue bars, *n* = 139 cells from six mice, *p* < 0.0001). Fluorescent intensities within EGFP-labeled hippocampal CA3 neurons were 38.92 ± 0.36 × 10^2^ in shRNA scramble control mice (red bar in Figure 6C, *n* = 81 cells from four mice) and 17.73 ± 0.48 × 10^2^ in *neuroligin-3* knockdown mice (blue bar, *n* = 52 cells from four mice, *p* < 0.0001). Thus, after *neuroligin-3* knockdown in the barrel cortex from associative learning mice, fewer hippocampal CA3 and CA1 neurons project their axons to the barrel cortex, and the activity strength of these hippocampal neurons significantly decreases. New axon projections from active hippocampal CA3 and CA1 neurons to barrel cortical neurons during associative learning are precluded by the knockdown of synapse linkage protein neuroligin-3. This result endorses the hypothesis that the secondary associative memory cells in hippocampal CA3/CA1 areas project their axons to innervate primary associative memory cells in the barrel cortex by the formation of new synapses [6].

### 2.4. The Knockdown of Neuroligin-3 in Hippocampal CA3 Attenuates the Connections from the Barrel Cortex to Hippocampal CA3, but Not from Hippocampal CA1 to CA3

We also examined the influence of *neuroligin-3* knockdown on the feedforward action by the connections from barrel cortical neurons to hippocampal CA1–3 neurons during associative memory using AAV2/retro-CMV-EGFP and AAV-DJ/8-U6-mNlgn3-EGFP or AAV-DJ/8-U6-mNlgn3-scramble-EGFP injected in the hippocampal CA3 area (white arrows in Figure 7B). The retrograde transportation from the axon terminals of hippocampal CA3 neurons to their somata was examined by confocal scanning of the barrel cortices and hippocampal CA1. Fluorescent intensities in the barrel cortices and hippocampal CA1 from *neuroligin*-3 knockdown mice appear lower than in those from shRNA scramble control. The relative fluorescent strengths of EGFP-labeled neurons in the barrel cortex at AP-1.70 mm were 0.05 ± 0.01 in shRNA scramble control mice (red bar in Figure 7C, *n* = 6) and 0.03 ± 0.001 in *neuroligin-3* knockdown mice (blue bar, *n* = 6, *p* = 0.2403). The relative strengths of EGFP-labeled neurons in the hippocampal CA1 area at AP-1.70 mm were 0.47 ± 0.08 in shRNA scramble control mice (red bar in Figure 7D, *n* = 6) and 0.48 ± 0.03 in *neuroligin-3* knockdown mice (blue bars, *n* = 6, *p* = 0.9372).

Confocal images at high magnification were also captured and analyzed. The number and the activity strength of EGFP-labeled barrel cortical neurons and hippocampal CA1 neurons appear substantially decreased in *neuroligin-3* knockdown mice (bottom panels in Figure 8A), compared with those in shRNA scramble control mice (top panels). The percentages of EGFP-labeled neurons in the barrel cortex were 50.47 ± 4.85 in shRNA scramble control mice (red bar in Figure 8B, *n* = 6) and 42.12 ± 8.6 in *neuroligin-3* knockdown mice (blue bar, *n* = 6, *p* = 0.8378). The percentages of EGFP-labeled neurons in the hippocampal CA1 area were 46.1 ± 5.03 in shRNA scramble control mice (red bar in Figur 8B, *n* = 6) and 38.25 ± 2.85 in *neuroligin-3* knockdown mice (blue bar, *n* = 6, *p* = 0.1843). Furthermore, fluorescent intensities in EGFP-labeled barrel cortical neurons were 12.41 ± 0.93 × 10^2^ in shRNA scramble control mice (red bar in Figure 8C, *n* = 61 cells from six mice) and 9.7 ± 0.51 × 10^2^ in *neuroligin-3* knockdown mice (blue bars, *n* = 79 from six mice, *p* = 0.0229). Fluorescent intensities within EGFP-labeled hippocampal CA1 neurons were 18.36 ± 0.7 × 10^2^ in shRNA scramble control mice (red bar in Figure 8C, *n* = 162 cells from six mice) and 13.57 ± 0.74 × 10^2^ in *neuroligin-3* knockdown mice (blue bar, *n* = 175 cells from six mice, *p* < 0.0001). Therefore, the activity strength of barrel cortical neurons weakens after the *neuroligin-3* knockdown in mice that have experienced associative learning, though the projection from the axons of barrel cortical neurons to the hippocampal CA3 area has not statistically changed. This result may be explained to be highly active in hippocampal CA3 neurons [6] that cannot be precluded by a knockdown of single molecules. This result endorses the hypothesis that the primary associative memory cells in sensory cortices strengthen their axon projections to innervate secondary associative memory cells in hippocampal CA3/CA1 areas by new synapse innervations [6].

## 3. Discussion

The new findings of our study are summarized as follows. After the formation of memories to associated signals that are the pair of whisker tactile and odorant stimulations, the synapse innervations from the hippocampal CA3 area to barrel cortices and the hippocampal CA1 area are strengthened (Figure 1 and Figure 2). The new synapse innervations from the barrel cortices and hippocampal CA1 area to the hippocampal CA3 area emerge (Figure 3 and Figure 4). Neuroligin-3 as a synapse linkage protein is required for the formation and strengthening of synapse interconnections (Figure 5, Figure 6, Figure 7 and Figure 8), which work for the positive interactions between primary associative memory cells in the barrel cortex and secondary associative memory cells in the hippocampus. Therefore, associative learning induces new synapse innervations and strengthens existing synapses (Figure 9) for the consolidation of associative memory. These experimental data obtained from an in vivo study support the hypothesis regarding the activity and strengthening of synapses [63,64], as well as reinforcing the view about their activity and interconnection [6].

It has been found that after associative learning, primary associative memory cells are recruited in sensory cortices [16,49,50,51,53] and secondary associative memory cells are recruited in the prefrontal cortex and the hippocampus [6,56,57]. Although the anterograde connection from those primary associative memory cells to those secondary associative memory cells is identified after associative memory forms [6,56], the feedback connection remains to be addressed [6]. The data presented in this study show the strengthened connection from the hippocampal CA3 area to the barrel cortex as well as the new synapse innervation from the barrel cortex to the hippocampal CA3 area (Figure 1 and Figure 2), which supports the hypothesis concerning the interconnections between primary and secondary associative memory cells [6].

In addition to the new and strengthened connections between associative memory cells, the functional state of these associative memory cells, or their spike-encoding capability, may be increased [6]. In previous studies, the functions of primary associative memory cells [50,53,55,65,66,67,68] and secondary associative memory cells [12,56,57,69] were increased after associative memory formed. In the present study, the expression of fluorescent proteins elevates after associative memory forms (Figure 2 and Figure 4). As the neuron activity strength is proportional to the expression of genes and proteins [6,70], the functional states of these associative memory cells are upregulated by their coactivity during associative memory. Such data endorse the hypothesis about activity-dependent positive recycling in the recruitment and refinement of associative memory cells [6], which works for the consolidation and maintenance of specific memory.

In terms of the cellular mechanisms underlying memory consolidation, the following possible mechanisms may be involved. The long-term potentiation of synaptic transmission has been thought of as a cellular mechanism for memory consolidation and maintenance [2,3,4,28,71]. In addition, the increased excitability of memory cells [50,53,55,65,66,68] and the increased interconnections among associative memory cells [67] after the formation of associative memory allow them to be driven and activated easily, which have also been thought to be relevant to memory consolidation. Furthermore, the increased interconnection among primary associative memory cells in sensory cortices and secondary associative memory cells in hippocampal CA3 areas in a positive feedback manner, as shown in the current study, may strengthen memory consolidation, which is supported by the observation that the hippocampal CA3 neurons possess higher excitability [72]. Therefore, multiple mechanisms are relevant to memory consolidation and maintenance.

Our study indicates that there are no synapse connections to hippocampal CA3 neurons from barrel cortical and hippocampal CA1 neurons under normal conditions (Figure 3 and Figure 4). This observation is consistent with those of previous studies [73,74,75,76]. However, new connections from barrel cortical neurons and hippocampal CA1 neurons to hippocampal CA3 neurons emerge during learning and associative memory. In addition to supporting a view of the activity and interconnection of synapses [6], this result implies that associative memory induces more interconnections among different brain regions for the enrichment of cognitive activities and emotional reactions [6].

## 4. Materials and Methods

### 4.1. The Selection of Animals

Experiments were done in accordance with the guidelines and regulations of the Administration Office of Laboratory Animals, Beijing China. All experimental protocols were approved by the Institutional Animal Care and Use Committee of the Administration Office of Laboratory Animals, Beijing China (B10831). C57BL/6JThy1-YFP mice (Jackson Lab. JAX stock #003782, Bar Harbor, ME, USA), the glutamatergic neurons in the cerebral brain of which were genetically labeled by yellow fluorescent protein (YFP), were used for all experiments [77,78]. Experimental mice were accommodated in a sterile barrier facility under the conditions of 12 h for day and night with sufficient food and water. The ambient temperature was 22 ± 2 °C. The relative humidity was 55 ± 5%. These conditions were set in the specific-pathogen-free (SPF) facilities. Mice 20 days postnatal with well-developed bodies were selected for training in associative learning with social stress. Mice were taken into the laboratory for one week to become familiar with the experimenters and the training apparatus.

### 4.2. The Study of Behavioral Tasks

The identification of associative memory formation was based on the fact that mice demonstrate odorant-induced whisker motion and whisking-induced olfactory responses after the simultaneous pairing of whisker and odor stimulations [16,53]. Mice on postnatal day 20 were divided into two groups. The whisker signal was 5 Hz mechanical stimulation and the odor stimulation was butyl acetate. One group of mice received the simultaneous pairing of odor and whisker stimulations, which was called the paired-stimulus group (PSG) or associative learning group. Another group of mice received unpaired whisker and odor stimulation, which was called the unpaired stimulus group (UPSG). The whisker stimulation and odor stimulation were given by the multiple sensory modal stimulator (MSMS) of a digital controller. The odorant stimulation was given by switching on a butyl-acetate-loaded tube and generating a small liquid drop in front of the noses of the mice. The intensity of butyl acetate odor is sufficient to activate olfactory bulb neurons, which were detected by two-photon cell imaging. The whisker stimulation was administered to the longer whiskers on the contralateral side of the barrel cortices where associative memory cells have been identified to reside. The intensity of whisker stimulation was sufficient to trigger whisker fluctuation, i.e., whisking-induced whisker motion. The parameters for whisker and odor stimulations for all mice were 20 s per training and five trainings per day with two-hour intervals for ten days. This training period was based on the fact that the onsets of odorant-induced whisker motion and whisking-induced olfactory response reach a plateau after approximately ten training days. The stimulus intensity, duration, and frequency were controlled by the MSMS and were fixed for each trial during the entire training period. For more information on the procedure, readers can explore other works by the authors’ group [6,16,53,55].

In the examination of whisker motion, whisker fluctuation magnitude, which is related to the absolute changes of whisking angles, was used [79]. The whisker motions of the mice in response to the odor test (butyl acetate toward the noses for 20 s) were recorded to quantify the onset time and strength of odorant-induced whisker motion, or associative memory. Odorant-induced whisker motion was accepted when whisker motions encountered the following criteria: the pattern of odorant-induced whisker motion was similar to that of the typical whisker motions induced by the whisker stimulation, but not different from the spontaneous whisking with lower magnitudes; and the whisking angle increased significantly compared with the angles seen in baseline controls and control groups. This odorant-induced whisker motion was originally induced by whisker stimulation, such that the odor signal evoked the recall of the whisker signal followed by the motion of longer whiskers similar to the innate reflex induced by the whisker stimulation, or conditioning responses [6,16,53,55].

The onset of whisking-induced olfactory responses, or the reciprocal retrieval of associative memory, was tested after training for ten days. In this test, the mice were placed in the central arm of a T-maze [80]. The whisking in their assigned whiskers was done by mechanical stimulation similar to the training paradigm. The mouse motions toward the other two arms were monitored in this test experiment. In this T-maze, each of the arms included either a block coated with butyl acetate or a block coated with glycerol. After whisker and odor signals were paired, the upregulation of olfactory sensitivity to this specific odorant could be measured by examining the mouse movement toward or away from the odorants in response to the whisker stimulation. Significant increases in moving into the arm with a preferential odorant or away from the arm with an unpleasant odorant indicated that the mice smelled the odorants. This upregulation of olfactory sensitivity during the whisking had the precondition of the pairing of whisker stimulation and odorant stimulation. This whisking-induced olfactory response implied the recall of an olfactory signal and the reflex of olfaction by whisker stimulus cue, or an associative memory [6]. The rates that mice correctly selected the arms were calculated from 10 repeats of the test for each mouse and averaged from the groups of mice to reduce variations. The onset of whisking-induced olfactory responses was warranted if the percentage of mice selecting the arm including the block without the butyl acetate coating significantly increased after associated training [16,53,55].

How basic units in memory traces interconnected and interacted with each other for associative memory was examined by using neural tracing to identify their connection and connective strength, in which AAV-carried fluorescent proteins were utilized in mice that experienced associative learning as well as formed associative memory. Retrograde neural tracing was performed by microinjecting AAV2/retro-CMV-EGFP (OBiO Inc., Shanghai, China) into either hippocampal CA3 areas or barrel cortices. After the associated signals were learned by pairing whisker and odorant signals and the associative memory had a few weeks to form, the retrograde distributions of fluorescent protein in barrel cortical neurons or hippocampal neurons were examined under confocal microscopy [6]. In order to isolate those neurons and axon projections newly formed by associative learning from innate neurons, C57 transgenic Thy-YFP mice in which excitatory neurons were genetically labeled by yellow fluorescent protein were used (YFP) [78]. The mRNA knockdown was performed by short-hairpin RNA (shRNA) specific for neuroligin-3 mRNA, which was carried by adeno-associated viruses (AAVs) and microinjected in this cortical area [58,59,60,61,62]. The microinjections of AAV-DJ/8-U6-mNlgn3-GFP (pAAV[shRNA]-U6-mNlgn3-EGFP) (VectorBuilder, Guangzhou, China) into the barrel cortex or hippocampal CA3 region were done two days before the training paradigm of associative learning. This approach was expected to deteriorate the expression of *neuroligin-3* in the neurons in these regions to prevent the formation and the linkage of new synapses based on neuroligin-3. The deterioration of synapse formation presumably impaired AAV uptake from axonal terminals and the retrograde transportation along with axons, leading to a decrease in and even loss of interconnections between the barrel cortex and the hippocampus. After the training paradigm of associative learning, those mice in the subgroups of neuroligin-3 knockdown by shRNA and shRNA-scramble with PSG were studied in terms of their behaviors in memory tasks and synapse innervations on the hippocampal neurons and/or barrel cortical neurons.

### 4.3. The Injections of Adeno-Associated Retro Viruses

Neural tracing to identify associative memory cells was performed by labeling presynaptic axons and axonal boutons with fluorescent proteins (FP) carried by Adeno-associated viruses (AAVs) [49,50,51,52,54]. In addition to tracing innate axons, this technique was particularly useful for labeling new synapse innervations from coactivated brain regions. For the retrograde tracing in the present study, AAV2/retro-CMV-EGFP was microinjected into brain regions that were thought of as the target areas of neuronal axon projection. These AAVs were taken up by the axonal terminals including innate axons and newly projected axons and then back-propagated through these axons toward neuronal somata for their expressions in the source areas of the neurons. The source fields of neuronal somata labeled by pAAV-EGFP, which were the target regions of pAAV-EPFG transportation, were scanned using a confocal microscope. Microinjections of AAV2/retro-CMV-EGFP were performed on mice at postnatal day 18, utilizing anesthesia induced by intraperitoneal injections of 4% chloral hydrate at a dosage of 400 mg/kg. The microinjections were conducted using a three-dimensional stereotaxic apparatus to precisely localize specific areas, such as the hippocampal CA3 area and the barrel cortex. Microinjection sites in barrel cortices were 1.7 mm posterior to the bregma, 2.75 mm lateral to midline, and 0.7 mm in depth, while the microinjection sites in the hippocampal CA3 area were located at 2.18 mm posterior to the bregma, 2.5 mm lateral to the midline, and were 1.8 mm in depth [81].

### 4.4. Neural Tracing by Adeno-Associated Retro Viruses

After the pAAV microinjections, a two-day recovery period was allotted for the mice. Following this, a consistent four-week training regimen was implemented to ensure adequate retrovirus expression in neurons. Subsequently, the areas of neuronal somata were scanned using a confocal microscope and analyzed through appropriate software tools. In order to clearly display three-dimensional images of new synapses in cortices, brain slices were placed into Sca/eA2 solution for 10 min to make them transparent [51,82]. The images of neuron soma areas were photographed under a confocal microscope using a 60× oil lens for high magnification and a 4× water-emerged lens for low magnification. A 466 nm excitation laser beam was used to activate AAV2/retro-CMV-EGFP. The emission wavelength used to scan AAV2/retro-CMV-EGFP was 522 nm. The resolution of the confocal imaging was 0.21 μm per pixel. The structures and quantity of the neurons and their florescent intensity were analyzed using the public-domain software package ImageJ 1.54 g and commercial software package Imaris 10.1 [6].

In the quantification of the retrograde neural tracing that showed axon projections and neuron somata, the neural tracing was compared between the source area where neuronal somata were located and the target area where axonal terminals are located. All fluorescence intensities are presented as relative values. Initially, the background of each image was removed using Image J. Subsequently, the relative fluorescence intensity of a defined region of interest (ROI) was measured. In the analysis of confocal images at low magnification, the multiplication of fluorescent intensity and area size were used as parameters showing the labeling of neurons and axons. In this way, the effects of different microinjection area sizes on the results between PSG and UPSG mice could be eliminated. In the comparisons, the differences between PSG and UPSG were calculated based on the division of the parameters in neuronal source areas by those in the axonal target area. The application of this relative alternation (relative folds in the *Y*-axis), which was also called the relative fluorescent strength of GFP-labeled neurons, could rule out the influence of microinjection area size and virus quantity on differences in results between PSG and UPSG [6]. In the analysis of confocal images at high magnification, the parameters used to show the labeling of neurons were the number of GFP-labeled neurons per slice section under the 60× magnification to indicate the number of associative memory cells as well as the average fluorescent intensity in each of EGFP-labeled neurons to show their activity strength after associative memory formed. In general, the strength of neuron activity was positively proportional to the expression levels of genes and proteins [6,70].

### 4.5. Statistical Analysis

The Mann–Whitney test was used for the statistical comparisons of experimental data of the intensity of fluorescents, the number of traced neurons, and the intensity of neuronal fluorescents in barrel cortices and hippocampal CA3–CA1 areas between the mice from the PSG and UPSG groups as well as between the mice from the neuroligin-3 knockdown and shRNA scramble control groups.

## Figures and Tables

**Figure 1 ijms-25-00711-f001:**
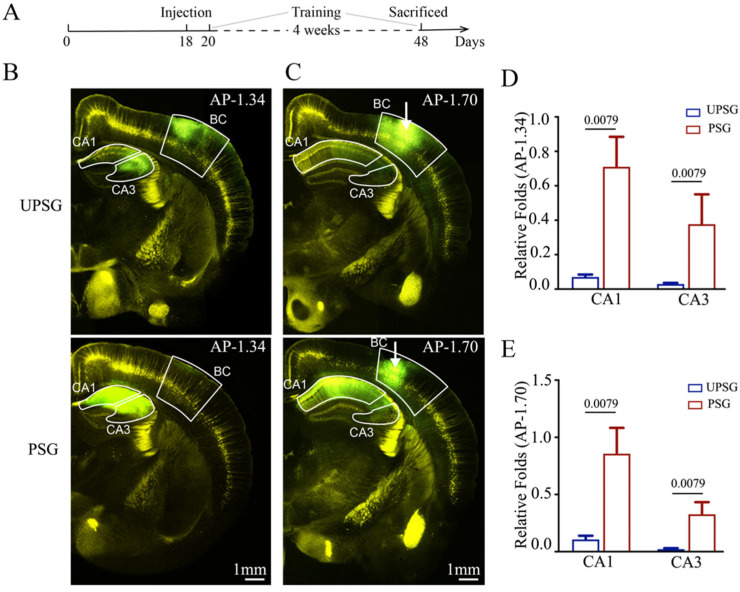
The neural connection from the hippocampus to the barrel cortex is strengthened after associative learning. Retro Viruses (AAV2/retro-CMV-EGFP) were microinjected into the barrel cortex (BC) of C-57 Thy-YFP mice two days before associative learning. After the training by paired or unpaired whisker and odor signals for four weeks, the brains of these mice were sliced and imaged by confocal microscopy. (**A**) Schematic illustration of the experimental timeline. (**B**,**C**) Images at low magnification from unpaired-stimulus group mice (UPSG, upper panel) and paired-stimulus group mice (PSG, lower panel), in which the photo images are taken at AP-1.34 mm (**B**) and AP-1.70 mm (**C**), respectively. White arrows indicate the microinjection sites in the BC. Yellow cells are YFP-labeled glutamatergic neurons. Green cells are EGFP-labeled neurons by pAAVs. Calibration bar represents 1 mm. (**D**,**E**) Relative fluorescent intensities of EGFP-labeled neurons in CA1 and CA3 of UPSG mice (blue bar) and PSG mice (red bar) measured at AP-1.34 mm (**D**) and AP-1.70 mm (**E**). The relative fluorescent intensity of the target brain region was normalized to the total fluorescent intensity at the microinjection site of AP-1.70 mm. The abbreviation AP refers to anterior–posterior. Statistics were performed by the Mann–Whitney test. Pooled data are presented as mean ± SEM.

**Figure 2 ijms-25-00711-f002:**
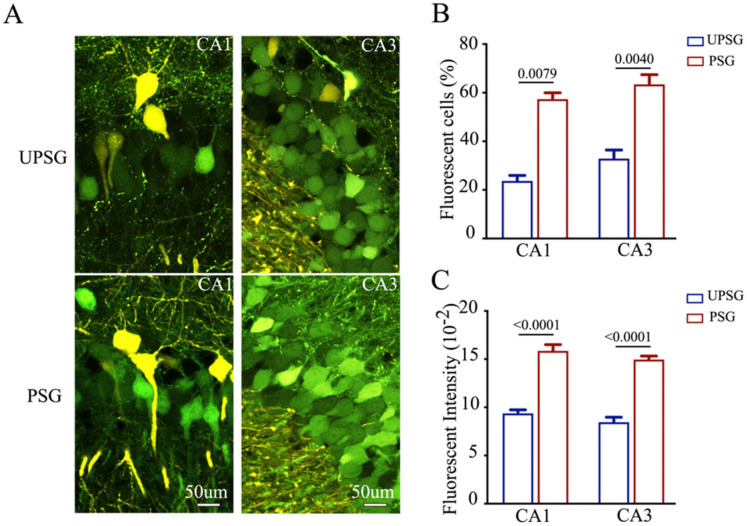
The number and activity intensity of hippocampal CA3 and CA1 neurons are increased after associative learning. (**A**) High magnification representative confocal images of hippocampal CA1 area (left panel) and CA3 area (right panel) from UPSG mice (upper panel) and PSG mice (lower panel) at AP-1.70 mm. Yellow cells are YFP-labeled glutamatergic neurons in the target areas. Green cells are EGFP-labeled neurons by pAAV-carried fluorescent genes. Calibration bar represents 50 μm. (**B**) The averaged percentage of EGFP-labeled neurons in CA1 and CA3 areas calculated and compared between UPSG mice (blue bar) and PSG mice (red bar). (**C**) Averaged fluorescent intensity of retro-traced green cells in CA1 and CA3.

**Figure 3 ijms-25-00711-f003:**
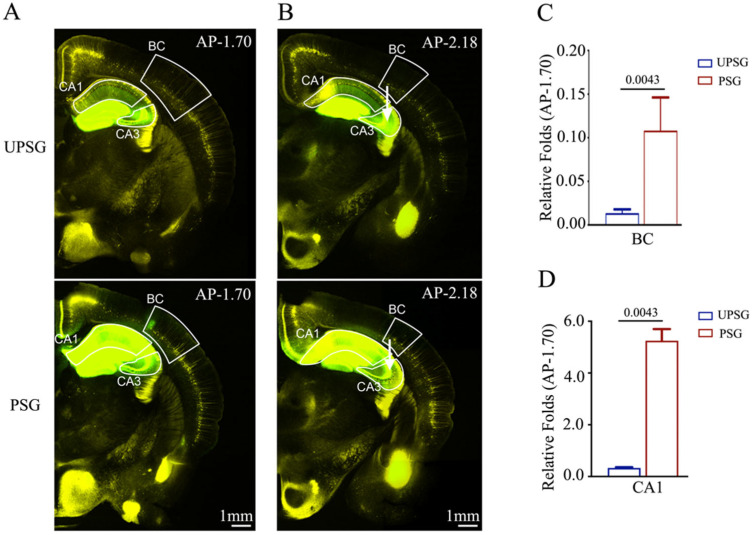
Neural connections from the barrel cortex to hippocampal CA3 and from hippocampal CA1 to CA3 are strengthened after associative learning. pAAVs were microinjected into the hippocampal CA3 area of C57 Thy-YFP mice two days before associative learning. After training by paired or unpaired whisker and odorant signals for four weeks, the brains of these mice were sliced and imaged by confocal microscopy. (**A**,**B**) Images at low magnification for UPSG mice (upper panel) and PSG mice (lower panel) photographed at AP-1.70 mm (**A**) and AP-2.18 mm (**B**). White arrows indicate microinjection sites in the hippocampal CA3 area. Yellow cells are YFP-labeled glutamatergic neurons. Green cells are EGFP-labeled neurons by pAAV-carried fluorescent genes. Calibration bar represents 1 mm. (**C**,**D**) Relative fluorescent intensities of EGFP-labeled neurons in the BC (**C**), hippocampal CA1 (**D**) of UPSG mice (blue bar) and PSG mice (red bar) measured and compared at AP-1.70 mm.

**Figure 4 ijms-25-00711-f004:**
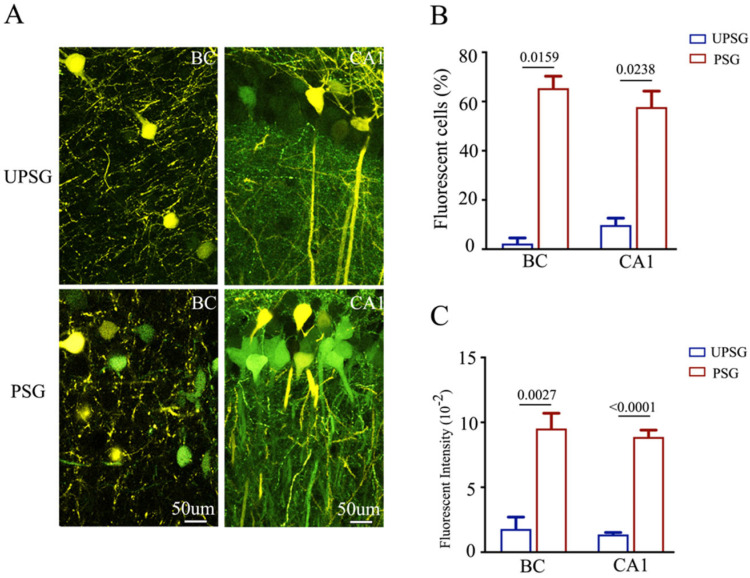
The number and activity intensity of barrel cortical and hippocampal CA1 neurons are increased after associative learning. (**A**) Representative confocal images of the BC (left panel) and hippocampal CA1 area (right panel) from UPSG (upper panel) and PSG (lower panel) mice at AP-1.70 mm under high magnification. Yellow cells are YFP-labeled glutamatergic neurons. Green cells are EGFP-labeled cells by pAAV-carried fluorescent genes. Calibration bar represents 50 μm. (**B**) Averaged percentage of EGFP-labeled neurons in the BC and hippocampal CA1 area calculated and compared between UPSG mice (blue bar) and PSG mice (red bar). (**C**) Averaged fluorescent intensity of retro-traced green cells in the BC and hippocampal CA1 area.

**Figure 5 ijms-25-00711-f005:**
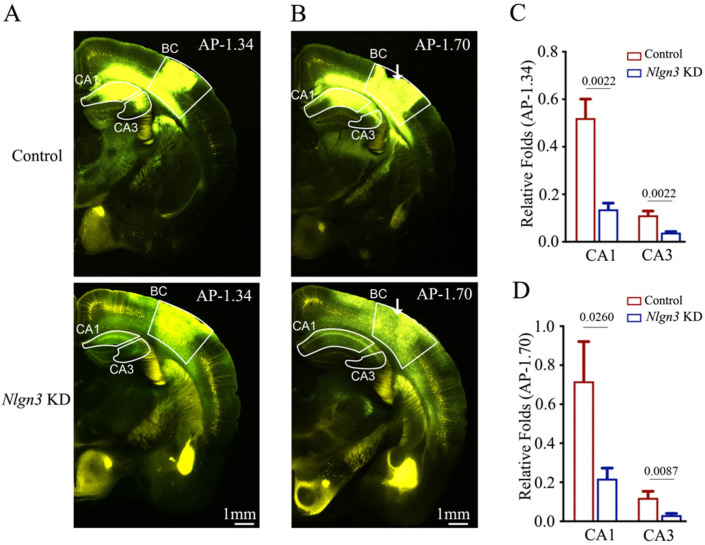
Neural connection from the hippocampus to the barrel cortex is weakened after neuroligin-3 knockdown in the barrel cortex. Retro Viruses (AAV2/retro-CMV-EGFP) and AAV-DJ/8-U6-mNlgn3-EGFP were microinjected into the barrel cortex (BC) of C57 Thy-YFP mice two days before associative learning. After training by paired whisker and odor signals for four weeks, the brains of these mice were sliced and imaged under a confocal microscope. (**A**,**B**) Images at low magnification from scramble control mice (upper panel) and Nlgn3 knockdown mice (Nlgn3 KD, lower panel), photographed at AP-1.34 mm (**A**) and AP-1.70 mm (**B**), respectively. White arrows indicate the microinjection sites in the BC. Yellow cells are YFP-labeled glutamatergic neurons. Green cells are EGFP-labeled neurons by pAAV-carried genes. Calibration bar represents 1 mm. (**C**,**D**) Relative fluorescent intensities of EGFP-labeled neurons in hippocampal CA1 and CA3 of scramble control mice (blue bar) and Nlgn3 KD mice (red bar) measured at AP-1.34 mm (**C**) and AP-1.70 mm (**D**).

**Figure 6 ijms-25-00711-f006:**
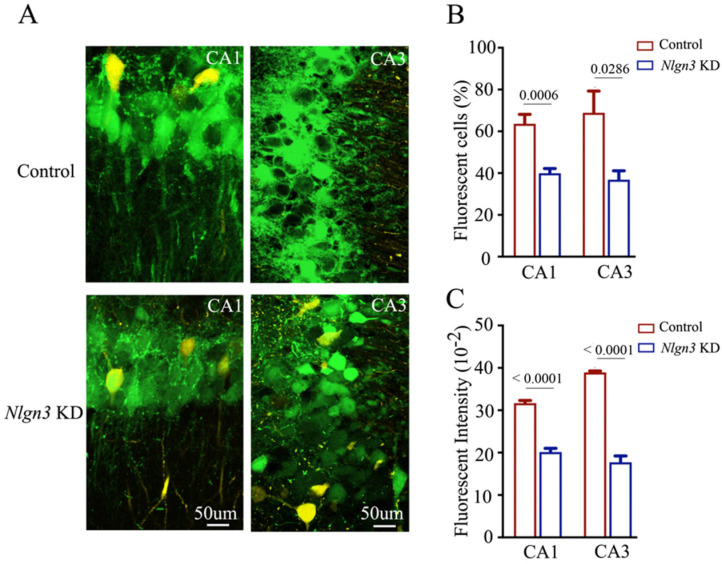
The number and activity intensity of hippocampal CA3 and CA1 neurons are decreased after neuroligin-3 knockdown in the barrel cortex. (**A**) Representative confocal images of hippocampal CA1 area (left panel) and CA3 area (right panel) from scramble control mice (upper panel) and Nlgn3 KD mice (lower panel) at AP-1.70 mm under high magnification. Yellow cells are YFP-labeled glutamatergic neurons. Green cells are EGFP-labeled neurons by pAAV-carried genes. Calibration bar represents 50 μm. (**B**) Averaged percentage of EGFP-labeled neurons in hippocampal CA1 and CA3 areas calculated and compared between scramble control mice (blue bar) and Nlgn3 KD mice (red bar). (**C**) Averaged fluorescent intensity of retro-traced green cells in hippocampal CA1 and CA3 areas.

**Figure 7 ijms-25-00711-f007:**
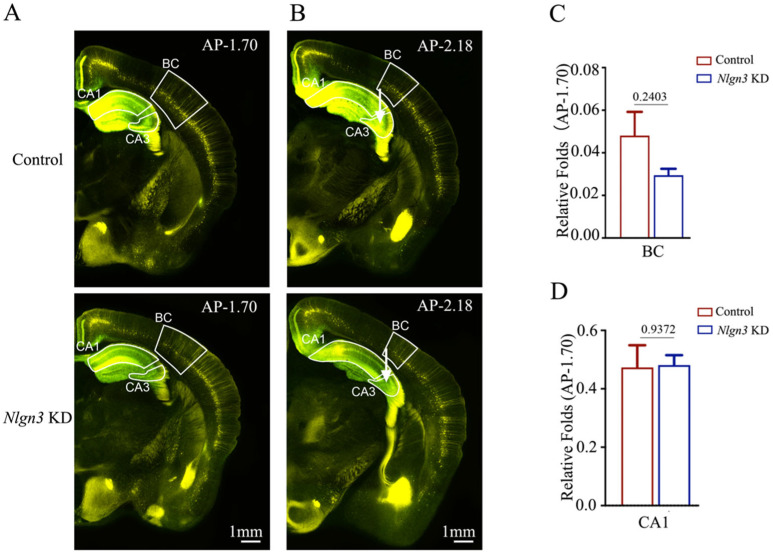
Neural connections from the barrel cortex to hippocampal CA3 and from hippocampal CA1 to CA3 seem unchanged after neuroligin-3 knockdown in CA3. Retro Viruses (AAV2/retro-CMV-EGFP) and AAV-DJ/8-U6-mNlgn3-EGFP were microinjected into the hippocampal CA3 area of C57 Thy-YFP mice two days before associative learning. After the training by paired whisker and odor signals for four weeks, the brains of these mice were sliced and imaged under a confocal microscope. (**A**,**B**) Images at low magnification for scramble control mice (upper panel) and Nlgn3 KD mice (lower panel), photographed at AP-1.70 mm (**A**) and AP-2.18 mm (**B**). White arrows indicate the microinjection sites in the hippocampal CA3 area. Yellow cells are YFP-labeled glutamatergic neurons. Green cells are EGFP-labeled neurons by pAAV-carried genes. Calibration bar represents 1 mm. (**C**,**D**) Relative fluorescent intensities of EGFP-labeled neurons in the BC (**C**), hippocampal CA1 area (**D**) of scramble control mice (blue bar) and Nlgn3 KD mice (red bar) measured and compared at AP-1.70 mm.

**Figure 8 ijms-25-00711-f008:**
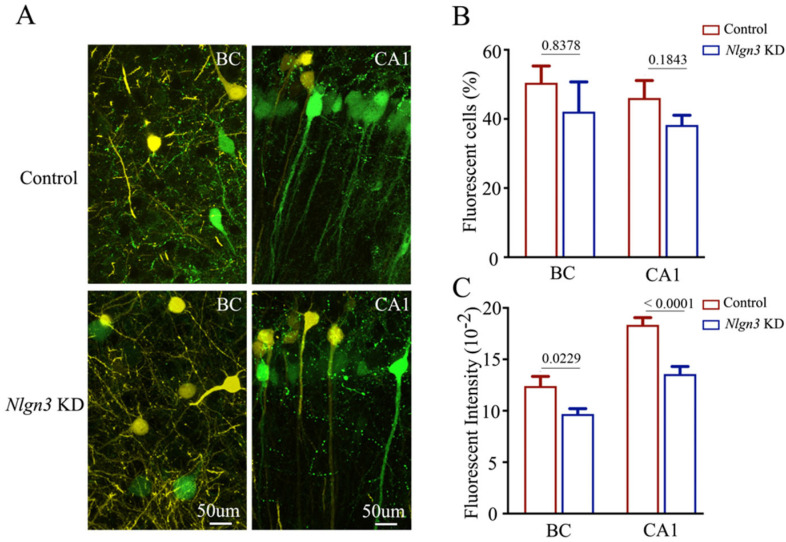
The number and activity intensity of barrel cortical and hippocampal CA1 neurons are decreased after neuroligin-3 knockdown in CA3. (**A**) Representative confocal images of the BC (left panel) and hippocampal CA1 area (right panel) from scramble control mice (upper panel) and Nlgn3 KD mice (lower panel) at AP-1.70 mm under high magnification. Yellow cells are YFP-labeled glutamatergic neurons. Green cells are EGFP-labeled cells by pAAV-carried genes. Calibration bar represents 50 μm. (**B**) Averaged percentage of EGFP-labeled neurons in the BC and hippocampal CA1 area calculated and compared between scramble control mice (blue bar) and Nlgn3 KD mice (red bar). (**C**) Averaged fluorescent intensity of retro-traced green cells in the BC and hippocampal CA1 area.

**Figure 9 ijms-25-00711-f009:**
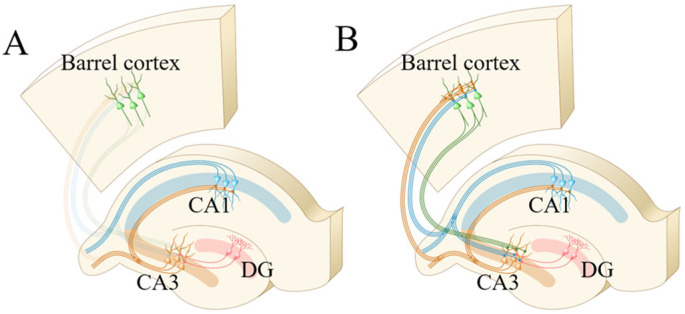
The interconnections between the hippocampal CA3 area and barrel cortex or hippocampal CA1 area are strengthened after associative memory. (**A**) Connections between the hippocampal CA3 area and the barrel cortex or hippocampal CA1 area before associative learning. (**B**) Connections between the hippocampal CA3 area and the barrel cortex or hippocampal CA1 area after associative memory forms.

## Data Availability

The data will be privded by the request from other researhers.

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
