# Peer review of "Neuroligin-3-Mediated Synapse Formation Strengthens Interactions between Hippocampus and Barrel Cortex in Associative Memory"

_ijms, 2024, doi:10.3390/ijms25020711_

Round 1
Reviewer 1 Report
Comments and Suggestions for Authors
I think it was a valuable discovery for the mechanisms of memory, which are still largely unknown, and a very beautiful experiment on the boundary between in vitro and in vivo. Having admitted this, I have a few requests.
The results should be of interest to many readers. Memory is definitely an area of great interest that biology is just barely reaching at present. With this in mind, please add more explanations. Most readers don't know much about markers. Most readers don't know much about markers, and may not know what to look for in a picture. The brain is very complex in parts. Not everyone knows where to cut and what to look for.
Often abbreviations are used without explanation.
L157 Please explain this: pAAV-CMV-EGFP-2A-MCS-3FLAG, pAAV-CMV-EGFP, AAV-DJ/8-U6-mNlgn3-GFP. Not all readers have expert knowledge of such markers. It should also be clear how and which manufacturer's products were used to create these constructs.
Overall, the report states that they were calculated using a t-test, but it is unclear how the image data was subjected to the t-test. For example, if images can be quantified in ImageJ, the fluorescence intensity itself should be a relative value. Multiple images should have been taken at different exposures and the amount of marker introduced into the cells should not be controllable. These comparisons require image normalisation. the authors must have done this in some way, how did they go about this process?
If the authors did not normalise, then they need to come up with some way to do it. For example, standardise the background image intensities (if normalisation is done correctly, the P-value should be even lower because of the reduced variability between the data).
Fig.1 Show the calculated P-value, not **. Because that is what the evidence is. Not how many or less, but the value.
Fig.1 I think this is a beautiful photograph of the sections, but for the benefit of the reader, please provide an illustration to assist the photograph. This paper deals with very interesting results and should be of interest to readers who are not researchers in this field. Which part is the hippocampus and where are CA1, CA3 and BA located respectively?
Fig.2 -Fig.7 Show the calculated P-values, not **.
Also, if intensity is used, standardisation is required by panelC, e.g.
Please explain how.
Fig.3 Here for the first time the concept of relative fold is mentioned, which was also mentioned in the introduction, please explain this in detail. Or this may be the desired form of standardisation, but the problem is that the ratios of something are often not normally distributed. If it is not normally distributed, it disqualifies the use of t-tests. It would be nice if the authors could show something like a normal QQ plot, or at least a histogram with supplements.
It is very fair and admirable that the authors cite so many other papers in the discussion. But the discussion should be about the data in this paper. Please specify more figures and explain why the results lead to them; the paragraphs from L527 lack this, for example. Especially the results in Fig. 7, for example.
It can be understood that the reason why the results were not positive for CA1 in Fig. 7 is Fig. 9A, which does not always have a connection here. Another experiment would help to ensure this,
Would it be possible to set up a PSG treatment zone? 
Reviewer 2 Report
Comments and Suggestions for Authors
Dear Authors,
I am sending you the comments on the Research Article titled “Neuroligin3-mediated synapse formation strengthens interactions between hippocampus and barrel cortex in associative memory”.
This manuscript reports the role of Neuroligin 3 in synaptic formation between the hippocampus and the barrel cortex during associative memory.
Generally, the methods used are sound and the data obtained are interesting. However, there are several points to be reconsidered:
- The section on materials and methods would be divided into sections indicating each technique or method separately.
- I would also add the anaesthesia and analgesia protocol used in stereotactic operations.
- I would add a diagram explaining the timelapse of the experiments.
- I would also separate the results into sections and remove the first paragraph of results because it is a repetition of the methods explained above.
- In Figure 1C PSG is oversized.
- I would indicate in white above photos CA1 and CA3.
- I am left wondering if the knockdown of the neuroligin 3 induces some kind of learning/memory dysfunction.
Best regards,
